# LBM-MHD Data-Driven Approach to Predict Rayleigh–Bénard Convective Heat Transfer by Levenberg–Marquardt Algorithm

Taasnim Ahmed Himika [1], Md Farhad Hasan [2,3], Md. Mamun Molla [4,5,*] and Md Amirul Islam Khan [6]

1   School of Science, Computing and Engineering Technologies, Swinburne University of Technology, Hawthorn, VIC 3122, Australia
2   Victoria State Government, Melbourne, VIC 3083, Australia
3   School of Computing, Engineering and Mathematical Sciences, La Trobe University, Melbourne, VIC 3086, Australia
4   Department of Mathematics & Physics, North South University, Dhaka 1229, Bangladesh
5   Center for Applied Scientific Computing (CASC), North South University, Dhaka 1229, Bangladesh
6   School of Civil Engineering, University of Leeds, Leeds LS2 9JT, UK
*   Correspondence: mamun.molla@northsouth.edu; Tel: +880-255668200 (ext.1519); Fax: +880-255668202

**Abstract:** This study aims to consider lattice Boltzmann method (LBM)–magnetohydrodynamics (MHD) data to develop  equations to predict the average rate of heat transfer quantitatively.  The present approach considers a 2D rectangular cavity with adiabatic side walls, and the bottom wall is heated while the top wall is kept cold. Rayleigh–Bénard (RB) convection was considered a heat-transfer phenomenon within the cavity. The Hartmann ($Ha$) number, by varying the inclination angle ($\theta$), was considered in developing the equations by considering the input parameters, namely, the Rayleigh ($Ra$) numbers, Darcy ($Da$) numbers, and porosity ($\epsilon$) of the cavity in different segments. Each segment considers a data-driven approach to calibrate the Levenberg–Marquardt (LM) algorithm, which is highly linked with the artificial neural network (ANN) machine learning method. Separate validations have been conducted in corresponding sections to showcase the accuracy of the equations. Overall, coefficients of determination ($R^2$) were found to be within 0.85 to 0.99.  The significant findings of this study present mathematical  equations  to predict the average Nusselt number ($\overline{Nu}$). The equations can be used to quantitatively predict the heat transfer without directly simulating LBM. In other words, the equations can be considered validations methods for any LBM-MHD model, which considers RB convection within the range of the parameters in each equation.

**Keywords:** lattice Boltzmann; Rayleigh–Bénard convection; magnetohydrodynamics; Levenberg–Marquardt algorithm; data-driven analysis; Nusselt number; Hartmann number; porosity; rectangular cavity

## 1. Introduction

The lattice Boltzmann method (LBM) is an efficient approach to investigate fluid flow through numerical simulations across different geometries at microscopic, mesoscopic, and macroscopic scales [1–8].  LBM is based on statistical mechanics and has immense potential to establish a data-driven analysis for scientific progress [9–11]. Therefore, high-dimensional nonlinear LBM data could be taken into account to calibrate any statistical model through high-performance computing (HPC). With the increasing demand for HPC, researchers have shifted their focus to fluid flow simulations by considering LBM across complicated grids [5,12–15]. LBM has been found to be efficient in flow simulations and heat transfer applications in hydrology, magnetohydrodynamics, and aerodynamics, to name a few [16–20]. Magnetohydrodynamics (MHDs) represents electrically conducting fluids in liquid metals and plasma flows.  The applications of MHD have been reported

in a wide range of applications, such as thermal engineering, geophysics, nuclear and hydroelectric power plants, astrophysics, and so on [18,21–24]. Therefore, the analysis of heat transfer in convective flow could be established by utilizing the LBM-MHD scheme.

The study of numerical heat transfer through different media is one of the popular fields of study among researchers [25–28]. Rayleigh–Bénard (RB) convection is one form of a phenomenon that takes place in a fluid layer assigned to a vertical temperature gradient and heated from the base [18,29–31]. The difference between buoyancy and gravity leads to fluid instabilities and convective electrical currents. This type of instability has been the subject of extensive research to identify a procedure to stabilize the system. One major reason could be the lack of understanding of the LBM data and correlations of the output with the input variables prior to the numerical modeling [32]. Therefore, it is necessary to determine the correlations to predict any upcoming phenomena linked to fluid instability. Several authors reported successful outcomes in stabilizing a system by applying an external magnetic field due to the induced electrical currents within the fluid [18,33,34]. However, most of the works were based on analysis through certain numerical parametric variations. Therefore, the prospect of establishing an LBM data-driven approach to determine correlations with the heat transfer prediction remains unnoticed.

Machine learning (ML) and deep learning (DL) are two important sectors of artificial intelligence (AI) with the ability for accurate data analysis and prediction model development [35–39]. With computational resources, ML and DL can build a multivariate model by taking high-dimensional non-linear data and developing correlations and numerical prediction models within different sectors. The model needs to be trained with a dataset to calibrate the model, and validations are performed through internal and independent datasets. However, the prediction model needs to be optimized through efficient training methods [40]. An inadequately optimized model will perform below the standard and yield noise within the model, leading to low correlation to predict the outcome. The Levenberg–Marquardt (LM) algorithm is one of the training methods for ML models, particularly for artificial neural networks (ANNs). LM develops the correlations by considering the input variables to provide a nonlinear least squares minimization (NLSM) solution. Therefore, it indicates that any numerically simulated data, including those from LBM, could be fed into the LM algorithm to understand the correlations among the variables through a quasi-ML approach along with numerical validations with the literature.

There is a shortage of literature on LBM data analysis through any efficient algorithm. However, some recent studies have reported the utilization of neural networks to optimize LBM data under the influence of MHD in natural convection. For example, Alqaed et al. [41] studied natural convection and entropy generation by applying a magnetic field with ANN and presented an equation based on the correlation development. However, the ML modeling equation lacks information on whether it can be used to predict total entropy across all geometries. In addition, the equation was not validated against any published literature to showcase the accuracy and robustness. Shah et al. [42] followed similar steps by adding radiation heat transfer. However, the validation methodology and the equations to predict the entropy remained ambiguous. On the other hand, the study presented by He et al. [43] was a much-improved one, as the correlation developments of LBM data were efficiently described through ANN and internal validations. Yet, the independent validations were still missed, and therefore, the accuracy of the correlations could not be expanded beyond the internal database.

This study aims to address the shortcomings within the literature by analyzing LBM data to establish correlations by considering the numerical variables, such as Rayleigh ($Ra$) number, Darcy ($Da$) number, Hartmann ($Ha$) number, inclination angle ($\theta$), and porosity ($\epsilon$), to predict the average rate of heat transfer ($\overline{Nu}$) by the LM algorithm. The obtained equations are presented in each section, including the statistical accuracy indicators, followed by validations within the literature in each step under various circumstances. The correlation coefficients ($R^2$) are found to be between 0.85 and 0.99, which provides more confidence in the accuracy of the numerical model.

## 2. Geometry of the Porous Cavity

The schematic diagram of the porous cavity along with associated coordinates is illustrated in Figure 1. The rectangular cavity in the 2D configuration includes the effect of a magnetic field to investigate the RB convection by considering incompressible and laminar fluid flow. The LBM data were extracted within these specifications. The cavity was assumed to be filled with electrically conducting fluid. $H$ denotes the vertical height, and the horizontal length is denoted by $L$, where $L = 2H$. Two vertical side walls were considered adiabatic, i.e., no heat transfer will occur. The top and bottom walls are cold and heated, represented by $T_c$ and $T_h$, respectively, where $T_h > T_c$. The LBM data were extracted through different parametric variations, such as the Rayleigh ($Ra$) number within the higher buoyancy range ($10^4$ and $10^5$). Three different Darcy ($Da$) numbers were considered, namely, $10^{-1}, 10^{-2}, 10^{-3}$, and Hartmann ($Ha$) numbers were considered to be between 0 and 100 to investigate the impact of the magnetic field. The impact of the magnetic field was further studied along with different inclination angles ($\theta$) ranging from 0 to 90. The porosity ($\epsilon$) was between 0.4 and 0.9. The gravitational acceleration ($g_y$) was acting downward. The uniform magnetic field was considered to be $B$ in Figure 1. The study assumes the Joule heating and viscous dissipation to be negligible to focus entirely on the impact of the magnetic field [18]. However, through the Boussinesq approximation, this particular assumption is validated, which ignores the density gradient, except from the appearance where the former is multiplied by $g_y$.

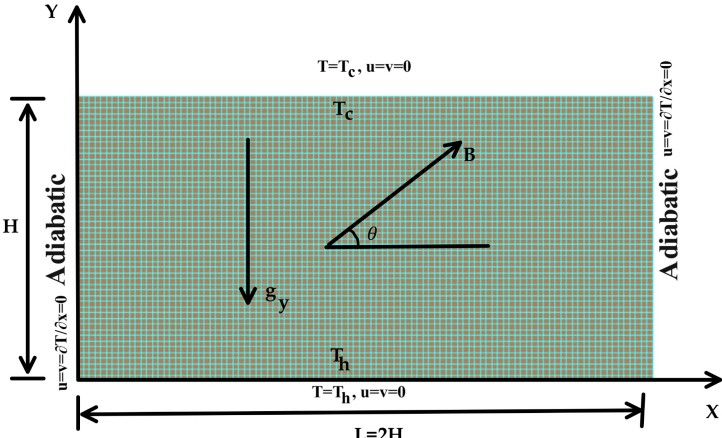

**Figure 1.** Considered geometry of the rectangular porous cavity along with the magnetic field.

## 3. Mathematical Formulations in Computation

### 3.1. Macroscopic Variables for Natural Convection in RB-MHD Flow in Porous Media

The formulation of LB equations using macroscopic governing equations is required to study MHD natural convection through porous media. These formulae include the energy equation, the Navier–Stokes equation with the Brinkman–Forchheimer model, and the continuity equation. However, these equations need to be converted to their non-dimensional form before being used to simulate MHD natural convection.

The dimensional equations for continuity, $u$-momentum, $v$-momentum, and energy are as follows:

$$\frac{\partial \bar{u}}{\partial \bar{x}} + \frac{\partial \bar{v}}{\partial \bar{y}} = 0 \tag{1}$$

$$\begin{aligned}
&\frac{\partial \bar{u}}{\partial \bar{t}} + \frac{1}{\epsilon}\left( \bar{u}\frac{\partial \bar{u}}{\partial \bar{x}} + \bar{v}\frac{\partial \bar{u}}{\partial y} \right) \\
&= \frac{1}{\rho}\left[ -\epsilon \frac{\partial \bar{p}}{\partial \bar{x}} + \bar{\mu}\left( \frac{\partial^2 \bar{u}}{\partial \bar{x}^2} + \frac{\partial^2 \bar{u}}{\partial \bar{y}^2} \right) + \sigma \epsilon B^2 (\bar{v}\sin\phi\cos\phi - \bar{u}\sin^2\phi) \right] - \frac{\epsilon v}{K}\bar{u}
\end{aligned} \tag{2}$$

$$
\frac{\partial \bar{v}}{\partial \bar{t}} + \frac{1}{\epsilon}\left( \bar{u}\frac{\partial \bar{v}}{\partial \bar{x}} + \bar{v}\frac{\partial \bar{v}}{\partial \bar{y}} \right)
$$

$$
= \frac{1}{\rho}\left[ -\epsilon\frac{\partial \bar{p}}{\partial \bar{y}} + \bar{\mu}\left( \frac{\partial^2 \bar{v}}{\partial \bar{x}^2} + \frac{\partial^2 \bar{v}}{\partial \bar{y}^2} \right) + g_y\epsilon\rho\beta(\bar{T} - T_c) + \epsilon\sigma B^2(\bar{u}\sin\phi\cos\phi - \bar{v}\cos^2\phi) \right] - \frac{\epsilon v}{K}\bar{v}
$$

$$
- \frac{1.75}{\sqrt{150\epsilon K}}|\mathbf{\bar{u}}|\bar{v} \tag{3}
$$

$$
\frac{\partial \bar{T}}{\partial \bar{t}} + \bar{u}\frac{\partial \bar{T}}{\partial \bar{x}} + \bar{v}\frac{\partial \bar{T}}{\partial \bar{y}} = \alpha\left( \frac{\partial^2 \bar{T}}{\partial \bar{x}^2} + \frac{\partial^2 \bar{T}}{\partial \bar{y}^2} \right) \tag{4}
$$

Meanwhile, the dimensionless governing equations could be written as the following:

$$
\frac{\partial u}{\partial x} + \frac{\partial v}{\partial y} = 0 \tag{5}
$$

$$
\frac{\partial u}{\partial t} + \frac{1}{\epsilon}\left( u\frac{\partial u}{\partial x} + v\frac{\partial u}{\partial y} \right)
$$

$$
= -\epsilon\frac{\partial p}{\partial x} + \frac{\Pr}{\sqrt{Ra}}\left( \frac{\partial^2 u}{\partial x^2} + \frac{\partial^2 v}{\partial y^2} \right) + \epsilon\frac{\Pr}{\sqrt{Ra}}\mathrm{Ha}^2(u\sin\phi\cos\phi - v\sin^2\phi) - \epsilon\frac{\Pr}{\sqrt{RaDa}}u \tag{6}
$$

$$
- \frac{1.75}{\sqrt{100\epsilon Da}}|\mathbf{u}|u
$$

$$
\frac{\partial v}{\partial t} + \frac{1}{\epsilon}\left( u\frac{\partial v}{\partial x} + v\frac{\partial v}{\partial y} \right)
$$

$$
= -\epsilon\frac{\partial p}{\partial y} + \frac{\Pr}{\sqrt{Ra}}\left( \frac{\partial^2 v}{\partial x^2} + \frac{\partial^2 v}{\partial y^2} \right) + \epsilon\theta\Pr + \epsilon\frac{\Pr}{\sqrt{Ra}}Ha^2(u\sin\phi\cos\phi - v\cos^2\phi) - \epsilon\frac{\Pr}{\sqrt{RaDa}}v \tag{7}
$$

$$
- \frac{1.75}{\sqrt{100\epsilon Da}}|\mathbf{u}|v
$$

$$
\frac{\partial T}{\partial t} + \bar{u}\frac{\partial T}{\partial x} + v\frac{\partial T}{\partial y} = \frac{1}{\sqrt{Ra}}\left( \frac{\partial^2 T}{\partial x^2} + \frac{\partial^2 T}{\partial y^2} \right) \tag{8}
$$

Here,
$\rho$ is the fluid density,
$\alpha$ is the thermal diffusivity,
$\epsilon$ is the porosity,
$T_c$ is the cold temperature,
$T_h$ is the hot temperature,
$\sigma$ is the electrical conductivity,
$\mu$ is the dynamic viscosity,
$H$ is the height of the cavity,
$B$ is the magnetic field strength,
$\phi$ is the angle of an applied magnetic field,
$\beta$ is the thermal expansion coefficient,
$g_y$ is the gravity acting downward along the y-axis,
$Da$ is the Darcy number,
$Ha$ is the Hartmann number,
$\Delta T = T_h - T_c$ is the temperature gradient between the top (hot) and bottom (cold) walls

$(T_h > T_c)$,
$|\mathbf{u}| = \sqrt{u^2 + v^2}$

The relations which are implied to convert the dimensional equations to non-dimensional form are

$$x = \frac{\bar{x}}{H}$$

$$y = \frac{\bar{y}}{H}$$

$$u = \frac{\bar{u}}{\left(\frac{\alpha}{H}\right)\sqrt{Ra}}$$

$$\nu = \frac{\mu}{\rho}$$

$$v = \frac{\bar{v}}{\left(\frac{\alpha}{H}\right)\sqrt{Ra}}$$

$$\theta = \frac{\bar{T} - T_c}{T_h - T_c}$$

$$t = \frac{\bar{t}}{\left(\frac{H^2}{\alpha}\right)}\sqrt{Ra} \quad (9)$$

$$p = \frac{\bar{p}}{p\left(\frac{\alpha}{H}\right)^2 Ra}$$

$$Ra = \frac{g_y \beta \Delta T H^3}{\nu \alpha}$$

$$Pr = \frac{\nu}{\alpha}$$

$$Da = \frac{K}{H^2}$$

$$Ha = BH\sqrt{\frac{\sigma}{\mu}}$$

### 3.2. LBEs for Heat Transfer and Fluid Flow

The lattice Boltzmann method is also referred to as thermal *LBM* or (*TLBM*) because it simulates the fluid flow mechanics by solving both the Boltzmann and the energy equations. *TLBM* calculates two distribution functions-$f_i$ for fluid field, and $g_i$ for temperature field. These distribution functions could be defined by considering the probability of particles in position $x$ at time $t$ moving toward each lattice direction $i$ with speed $c_i$ during time $\Delta t$. It enables the formulation of macroscopic fluid parameters, such as pressure, temperature, and velocity. In addition, the fluid domain is discretized into homogeneous lattice nodes. The inclusion of the *BGK* approximation into the LB equation results in the following equations with an external force [18]:

For the flow field:

$$f_i(\bar{x} + \bar{e}_i \Delta t, t + \Delta t) = f_i(\bar{x}, t) - \frac{f_i(\bar{x}, t) - f_i^{eq}(\bar{x}, t)}{\tau_v} + \Delta t \bar{F}_i \quad (10)$$

For the temperature field:

$$g_i(\bar{x} + \bar{e}_i \Delta t, t + \Delta t) = g_i(\bar{x}, t) - \frac{g_i(\bar{x}, t) - g_i^{eq}(\bar{x}, t)}{\tau_\alpha} \quad (11)$$

Here, $\tau_v = 3\nu + 0.5$, and $\tau_\alpha = 3\alpha + 0.5$ are the single-relaxation times (SRTs) that define the approaching rate to the equilibrium state. Meanwhile, kinematic viscosity $\nu$ and thermal diffusivity $\alpha$ are presented as the following:

$$\nu = \left(\tau_v - \frac{1}{2}\right)c_s^2 \Delta t \tag{12}$$

$$\alpha = \left(\tau_c - \frac{1}{2}\right)c_s^2 \Delta t \tag{13}$$

where $c_s$ is the speed of sound, $c_s = c/\sqrt{3}$, and $c$ is the spacing among the lattice.

The external force term $\bar{F}_i$ consists of three forces: $F^M{}_i$ (for MHD), $F^P{}_i$ (for porous media, which is the Brinkman–Forcheimer model), and finally, $F^b{}_i$ (buoyancy term):

$$F_i = F^M{}_i + F^P{}_i + F^b{}_i \tag{14}$$

On the other hand, magnetic force $F^M{}_i$ acts in $x$ and $y$ directions, and is expressed as the following [44]:

$$F^M{}_i = F_x + F_y \tag{15}$$

$$F_x = \frac{Ha^2\mu}{L^2}(v\sin\phi\cos\phi - u\sin^2\phi) \tag{16}$$

$$F_y = \frac{Ha^2\mu}{L^2}(u\sin\phi\cos\phi - v\sin^2\phi) \tag{17}$$

The buoyancy force term can be expressed as

$$F^b{}_i = \rho g_y \beta (T_h - T_c) \tag{18}$$

The applied magnetic field does affect the force term. The present study considers the external magnetic field is applied in different directions. The direction is horizontal, vertical, or in other angles (for example, $\theta = 0, 45, 90$). The external *MHD* forces acting in $x$ and $y$ directions are presented as the following:

$$F_x = 3\omega_k \rho \epsilon A[(v\sin\theta\cos\theta) - (u\sin^2\theta)] \tag{19}$$

$$F_y = 3\omega_k \rho \epsilon (g_y \beta (T - T_{ref})) + A[(u\sin\theta\cos\theta) - (v\cos^2\theta)] \tag{20}$$

The magnetic buoyancy force in terms of weighting factor is written as

$$F^b{}_i = 3\omega_k \rho \epsilon g_y \beta (T - T_m) \tag{21}$$

Here, $T_m = (T_h + T_c)/2$.

The body force for porous media, $F^P{}_i$, is expressed through Ergun's equation [45]:

$$\bar{F}_i = -\frac{\epsilon v_k}{K}\bar{u} - \frac{\epsilon F_\epsilon}{\sqrt{K}}|\bar{u}|\bar{u} + \epsilon \bar{G} \tag{22}$$

where, $F_\epsilon$ represents the geometric function ($F_\epsilon = \frac{1.75}{\sqrt{150}}$), $K$ is the permeability ($K = Da \cdot H^2$) with $H$ symbolizing the domain height, $v_k$ represents the kinematic viscosity, and $\bar{G}$ represents the external body force term.

An alternative equation of force term, $F^P{}_i$, for porous media was proposed by Mohamad [46] to obtain the accurate solution of hydrodynamics, which is the following:

$$F^P{}_i = -w_k \left[9\left(\frac{\epsilon v}{K}\right)(ue_x + ve_y) + \frac{F_\epsilon \epsilon}{\sqrt{K}}(|\bar{u}|ue_x + |\bar{v}|ve_y)\right] \tag{23}$$

The present study considers *D2Q9*, i.e., two-dimensional nine-velocities, model [47]. Therefore, the equilibrium distribution functions ($f_i^{eq}$) for the *D2Q9* model of porous media is written as the following:

$$f_i^{eq} = \omega_k \rho \left[ 1 + \frac{\bar{e}_i . \bar{u}}{c_s^2} + \frac{(\bar{e}_i . \bar{u})^2}{2\epsilon c_s^4} - \frac{|\bar{u}|^2}{2\epsilon c_s^2} \right] \tag{24}$$

Here, $\epsilon$ is denoted as the porosity. The discrete velocities $\bar{e}_i$ for the *D2Q9* model have different parametric values and are expressed as mentioned by [48]:

$$\bar{e}_i = \begin{cases} (0,0) & \text{for i} = 0 \\ cos[(i-1)\pi/4], sin[(i-1)\pi/4] & \text{for i} = 1\text{–}4 \\ \sqrt{2}(cos[(i-1)\pi/4 + \pi/4], sin[(i-5)\pi/2 + \pi/4]) & \text{for i} = 5\text{–}8 \end{cases} \tag{25}$$

The values of the weighting factor $\omega_k$ are the following:

$$\omega_k = \begin{cases} 4/9 & \text{for i} = 0 \\ 1/9 & \text{for i} = 1\text{–}4 \\ 1/36 & \text{for i} = 5\text{–}8 \end{cases} \tag{26}$$

The thermal equilibrium energy function can be expressed as the following [48]:

$$g_i^{eq} = \omega_k T \left[ 1 + \frac{1}{c_s^2} \bar{e}_i . \bar{u} \right] \tag{27}$$

### 3.3. Boundary Conditions

Boundary conditions were defined for the four walls of the rectangular cavity for the purpose of simulation. Boundary conditions are generally described as distribution functions (DFs) in LBM. It is required to determine the DFs at the boundary nodes according to the macroscopic conditions. The procedure is attributed with ensuring the stability and accuracy of the mathematical model [49].

#### 3.3.1. Boundary Conditions for Fluid Flow

The no-slip (also known as bounce-back) boundary condition was applied on the walls of the rectangular cavity. As an aftermath of the particles' collision, the outgoing DF goes in the reverse direction of the incoming DF at a particular position within the boundary. The following expressions represent the boundary conditions of this study:

At right wall: $f_{3,m} = f_{1,m}$, $f_{7,m} = f_{5,m}$, and $f_{6,m} = f_{8,m}$

At left wall: $f_1 = f_3$, $f_5 = f_7$ and $f_8 = f_6$

At top wall: $f_{4,n} = f_{2,n}$, $f_{8,n} = f_{6,n}$ and $f_{7,n} = f_{5,n}$

At bottom wall: $f_2 = f_4$, $f_5 = f_7$ and $f_6 = f_8$

where $m$ and $n$ represent the domain's lattice for length and height, respectively.

#### 3.3.2. Thermal Boundary Conditions

As described earlier, the top ($T_c$) and the bottom walls ($T_h$) have constant temperatures, but they have different values. The other walls are adiabatic and, therefore, are not participating in the mass transfer.

At isothermal cold top wall: $g_{4,n} = -g_{2,n}$

At isothermal hot bottom wall: $g_2 = T_{wall} (w_2 + w_4) - g_4$

Here, $T_w$ is used for the 2nd-order Zou-He boundary conditions

At adiabatic west wall: $g_{i,0} = g_{i,1}$, for i = 1–8

At adiabatic east wall: $g_{i,m} = g_{i,m-1}$, for i = 1–8

where, $m$ and $m-1$ are the boundary lattice and the lattice inside the enclosure near the boundary, respectively.

### 3.4. Rate of Heat Transfer

In the numerical investigation of the convective heat transfer problem, the Nusselt number ($Nu$) is an important parameter. The $Nu$ number describes the heat transfer rate

due to temperature gradient. The local $Nu$ number at hot walls and the average $Nu$ number ($Nu_{avg}$) calculated for the entire domain are formulated as the following [47,50]:

$$Nu(x) = -\frac{H}{\Delta T}\frac{\partial \overline{T}}{\partial \overline{y}} \tag{28}$$

$$Nu_{avg} = \frac{1}{H}\int_0^H Nu(x).dx \tag{29}$$

where $L$ denotes the length of the cavity.

Clever and Busse [51] defined a modified formulation for the average Nusselt number, $\bar{N}u$ in their work, and it is written as

$$\overline{Nu} = 1 + \frac{<\bar{v}\cdot\bar{T}>}{\sqrt{Ra}\Delta T\alpha/H} \tag{30}$$

where $H$ represents the distance between the bottom and top walls, $\Delta T$ is difference between temperature of top and bottom walls, $<\cdot>$ shows the average over whole flow domain, and $v$ denotes the velocity component of the vertical direction.

For the RB convection, He at al. [52] formulated an equation for the average Nusselt number $\overline{Nu}$ in terms of critical Rayleigh $Ra_c$ and Rayleigh number $Ra$:

$$\bar{N}u_{EM} = 1.56 \times (Ra/\mathrm{Ra}_c)^{0.296} \tag{31}$$

where $Ra_c = 1707.06$.

*3.5. LM Algorithm*

The LBM-MHD-RB data-driven work in this study is analyzed by the LM algorithm. It is a hybrid method that considers both the Gauss–Newton and steepest descent approaches for the convergence criteria to reach an optimal solution. There is an inherent trade-off between Gauss–Newton and the steepest descent based on the requirements of the problem. For instance, if Gauss–Newton alone cannot solve a problem, the LM algorithm links the steepest descent approach for traversing the design space and determining the optimal solution. This technique is most effective in solving non-linear equations. The correlations to predict the output parameter by considering the influential parameters of LBM are typically non-linear, and therefore, the LM algorithm was a suitable option for the surface analyses.

LM develops the trust region for different computational applications. In the LM method, the difference in the weights ($w_i$) is obtained by determining the following [53]:

$$\alpha'\Delta = -1/2(\nabla)\lambda \tag{32}$$

where $\alpha'$ is the matrix of the optimization field, and $\lambda$ is the mean-squared network error. The term $\lambda$ is achieved by the following equation [53]:

$$\lambda = 1/N\sum_{k=1}^N [\vec{q}(x_k) - \vec{d}_k]^2 \tag{33}$$

where $N$ is the number of examples; $\vec{q}(x_k)$ is the output of the network aligning with the example $x_k$; and $\vec{d}_k$ is the expected outcome.

Finally, the matrix $\alpha'$ elements are obtained by the following [53]:

$$\alpha'_{ij} = (1 + \zeta\delta_{ij})\sum_{r=1}^z\sum_{k=1}^N\left[\frac{\partial y_r(x_k)}{\partial w_i}\frac{\partial y_r(x_k)}{\partial w_j}\right] \tag{34}$$

where $z$ is the number of the desired output from the network.

At the commencement of the algorithm, $\alpha'$ and $\nabla(\lambda)$ are a major part of the evaluation, followed by the obtained solution on $w_i$. The LBM-obtained data were initially analyzed

in the R programming environment using library packages *"dplyr"* [54], followed by *"pracma"* [55] to optimise the matrix $\alpha$. The analyzed, clean dataset was fed into OriginPro to perform the non-linear surface analyses. In each step of iteration, the $\lambda$ value was calculated through the model, and the iteration was not terminated unless an optimal solution was reached. The convergence criteria were set if a coefficient of determination ($R^2$) was reached above 0.8 for any specific function. A separate function was chosen for the iteration if the required $R^2$ was not obtained despite adjusting the iteration cycle $\zeta$. It should be mentioned here that the present study does not aim to perform an AI approach, such as neural networks, due to the limited data for the analysis, and since the solution was reached by the LM algorithm by solving the non-linear least squares curve fitting, the process was terminated once a standard $R^2$ was achieved, subject to further validations. Therefore, once the solution at $w_i$ was reached, the equation was obtained to validate the accuracy. The obtained equation was initially checked through the interpolated dataset, which was at least 3000, depending on the percentage of outliers. The fundamental aim was to always have $R^2$ more than 0.95, and hence the outlier detection test was conducted on the obtained dataset. The initial process is known as the LM iteration/learning cycle, with each step destined to reduce the error from the previous one. The $\zeta$ parameter is an adjustment at each cycle. The process keeps on running unless a good correlation coefficient is reached. Once the iteration was terminated, the equation was obtained and immediately tested for the accuracy.

### 3.6. Code Convergence Criteria

In the LBM simulations, all the computations were terminated when the velocity field, as well as temperature, reached the following convergence criteria:

$$\frac{\sum |\psi^{n+1} - \psi^n|}{\sum |\psi^{n+1}|} < 10^{-15} \tag{35}$$

where $\psi$ is either the velocity $u$ or the temperature $T$, $n$ represents the iteration index, and finally, the summation was applied over the whole domain of interest.

Meanwhile, the LM algorithm followed an iterative method unless an accurate correlation coefficient was obtained as described in Figure 2.

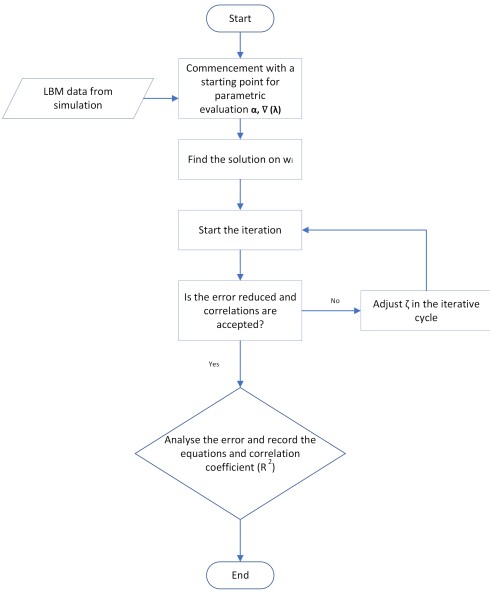

**Figure 2.** Flowchart of LM algorithm using LBM data for the correlation development.

## 4. Materials and Methods

LBM simulations for MHD-RB convection were performed in Fortran 90 [56] by using Microsoft Visual Studio Code$^{TM}$. Boundary conditions, collision operators, streaming functionality, and convergence criteria were all included as subroutines. The base code considered all those subroutines for the iterations. The iterations continued until the convergence criteria were obtained. The computations were performed on a Windows 10 computer in an 11th Gen Intel(R) Core$^{TM}$ i9 2.60 GHz processor with 64 GB RAM. The streamlines and isotherms were visualized by using Tecplot 360 2022 R1 version (https://www.tecplot.com).

As mentioned earlier, the LM algorithm was performed through the R programming language [57] by RStudio$^{TM}$ open source software using library packages *pracma* [55], followed by data organising, equations validation, and correlation development in OriginPro [58]. The library package *pracma* determines a large number of functions from the numerical analysis for any math function. Prior to that, the popular *dplyr* [54] package was used for data manipulation and visualization. It should be mentioned here that the same computer was utilized for LM algorithm development, which was also used for the LBM simulations. However, both RStudio and OriginPro were operated with NVIDIA RTX A3000 GPU power for fast implementation of the model optimization and correlations development.

## 5. Results

The primary purpose of the present study is to develop the correlation among the important variables to quantitatively predict $\overline{Nu}$ number, which is representative of the average heat transfer rate. Therefore, the results will discuss the numerical correlations based on LBM-MHD data interpretation through the LM algorithm. Each segment demonstrates the obtained outcome from the non-linear surface analysis, followed by validations with literature to showcase the accuracy of the obtained equations. However, two separate comprehensive analyses are first conducted to pinpoint some of the significant changes in the streamlines and isotherms.

### 5.1. Effect of Numerical Parameters on Streamlines

The impact of *Ra* and *Ha* numbers, as well as $\epsilon$ on streamlines, is illustrated in Figure 3 under a constant $\theta = 0$. The combined analysis will depict each variable's influence on the streamlines' pattern.

Figure 3a is assigned with $Ra = 10^5$, $Da = 10^-2$, $\epsilon = 0.4$, and $Ha = 0$, leaving entirely no impact of the external magnetic field. As per Figure 3a, three separate circular rolls distributed within a symmetry within the cavity were observed. However, the circular rolls in the left and right locations of the cavity exhibited almost similar characteristics with the maximum contour values at the center. However, the circular roll in the middle demonstrated the opposite and negative contour values. This behavior could be attributed to side-heated adiabatic walls and the top and bottom walls being active in the heat transfer process. Therefore, the circular roll in the middle directly depicted the effect of convective characteristics instead of conductive ones [18]. However, as the *Ha* number increased from 0 to 50, the shape and contour values changed significantly, as seen in Figure 3b. The Bénard cell reduced from 3 to 1 and started to stretch from the central region. The maximum contour value also reduced from 8 to 6, which is almost a 25% reduction due to the 50% augmentation in the *Ha* number. Therefore, it was expected that increasing *Ha* number would keep on reducing the heat conduction. The hypothesis was confirmed from Figure 3c, where the maximum contour value at the center plummeted to 1. By increasing the *Ha* number from 50 to 100, the maximum contour value reduced by approximately 83.33%, indicating the negative influence of the external magnetic field and the existence of restriction within the cavity to reduce the heat transfer mobility.

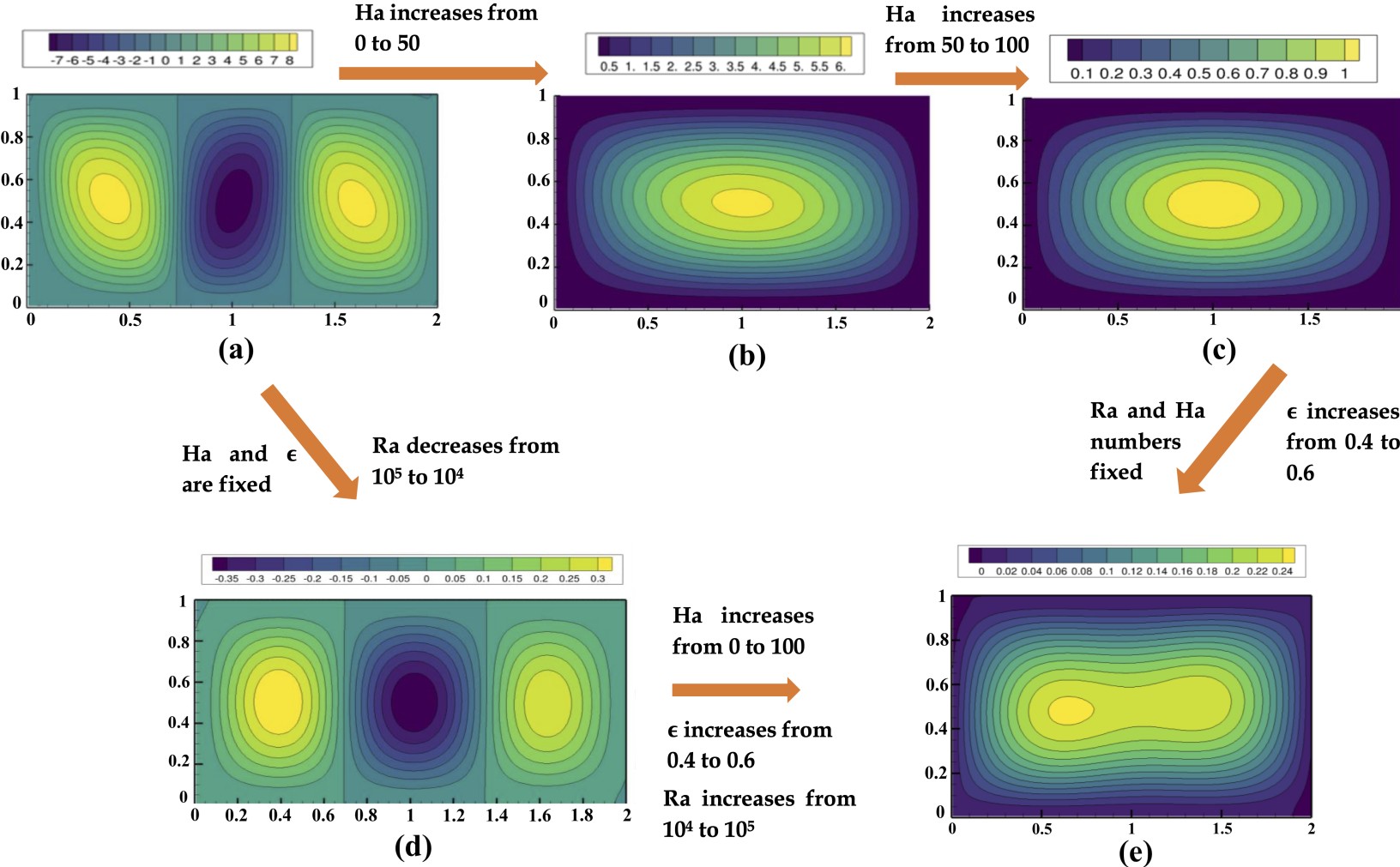

**Figure 3.** Illustrations of streamlines as $Ra$, $Ha$ numbers, and $\epsilon$ varied under $\theta = 0$, and $Da = 10^{-2}$: (**a**) $Ra = 10^5$, $\epsilon = 0.4$, and $Ha = 0$, (**b**) $Ra = 10^5$, $\epsilon = 0.4$, and $Ha = 50$, (**c**) $Ra = 10^5$, $\epsilon = 0.4$, and $Ha = 100$, (**d**) $Ra = 10^4$, $\epsilon = 0.4$, and $Ha = 0$, (**e**) $Ra = 10^5$, $\epsilon = 0.6$, and $Ha = 100$.

In the second part of this analysis, the influence of $Ra$ numbers and $\epsilon$ was observed. By comparing Figure 3a and Figure 3d, the impact of the $Ra$ numbers could be observed by keeping $Ha$ and $\epsilon$ constant. As $Ra$ decreased from $10^5$ to $10^4$, the maximum contour value decreased from 8 to 0.3, which is a rapid 96.25% reduction. With this observation, the impact of buoyancy in the RB convection could be understood. In the next part, all the variables, namely $Ra$ and $Ha$ numbers, and $\epsilon$ were increased concurrently as shown in Figure 3e. According to Figure 3e, three Bénard cells reduced to one but demonstrated strong attraction toward the thermal walls by showing similarity with the thermal dipole. The maximum contour value was 20% reduced from 0.3 to 0.24, and no negative value was recorded. While the increased Ha number was directly responsible for negative contour values, the increased $\epsilon$ and $Ra$ numbers enhanced the heat transfer phenomena, leading to 0 as the lowest contour value within the cavity. The impact of $\epsilon$ was also tested by keeping $Ha$ and $Ra$ numbers constant. By comparing Figure 3c,e, the influence of $\epsilon$ could be analyzed, where the maximum contour value decreased from 1 to 0.24. However, in both Figure 3c,e, the $Ha$ number is 100, which repelled the heat transfer application. Therefore, the contour value reduced significantly by about 140%.

*5.2. Isothermal Changes*

The final section of the results focuses on the changes in the isotherms, similar to the previous section. Figure 4 represents such changes in five different frames.

The impact of $Ha$ can be observed from Figure 4a–c by increasing from 0 to 100 in three separate frames. The distribution of isotherms is kept within 0 to 1. As per Figure 4a, the isothermal lines demonstrate an oscillating pattern due to the heat transfer within the cavity without the influence of $Ha$ number. The pattern within 1 to 1.5 of the horizontal axis is the opposite of what was observed within 0 to 1 of the same axis. This behavior could be linked with the conduction and convective rolls observed in Figure 3a, where the middle convective rolls represent the negative contour values. Therefore, the pattern of the isotherm from 1 to 1.5 on the horizontal axis is the opposite. As the $Ha$ number increases from 0 to 50, the isothermal lines exhibit uniformity within the cavity, as the oscillation disappears and all the lines start to become quasi-linear as seen in Figure 4b. The presence of the $Ha$ number leads to the presence of a magnetic field which develops the Lorentz force within the cavity. Therefore, the instability within the thermal walls is reduced. Further decreasing oscillation could be observed from Figure 4c, where the isothermal lines edge closer to the linearity. While a wavy pattern could be seen at the lowest contour, the isothermal lines are quite linear at the maximum contour values, which are closer to the horizontal axis.

Meanwhile, the effect of plummeting $Ra$ numbers could be observed from Figure 4d, where decreasing $Ra$ from $10^5$ to $10^4$ significantly impacts the isothermal patterns. It could be observed that due to the decreased buoyancy, the isothermal lines show minor oscillation with a minimal peak in each line. The isothermal line close to the horizontal axis show a linear pattern due to the lack of buoyancy strength within the cavity. However, as $Ha$ is increased from 0 to 100, $Ra$ is increased from $10^4$ to $10^5$, and finally, $\epsilon$ is also increased from 0.4 to 0.6. The isothermal lines are almost linear throughout the cavity due to the strong influence of the $Ha$ number in particular as seen in Figure 4e. In fact, keeping $Ha = 100$ constant and increasing $\epsilon$ from 0.4 to 0.6 does not significantly impact the isothermal lines either, due to the existence of the Lorentz force. By comparing Figure 4c,e, the impact of the $Ha$ number in the RB convection could be well understood.

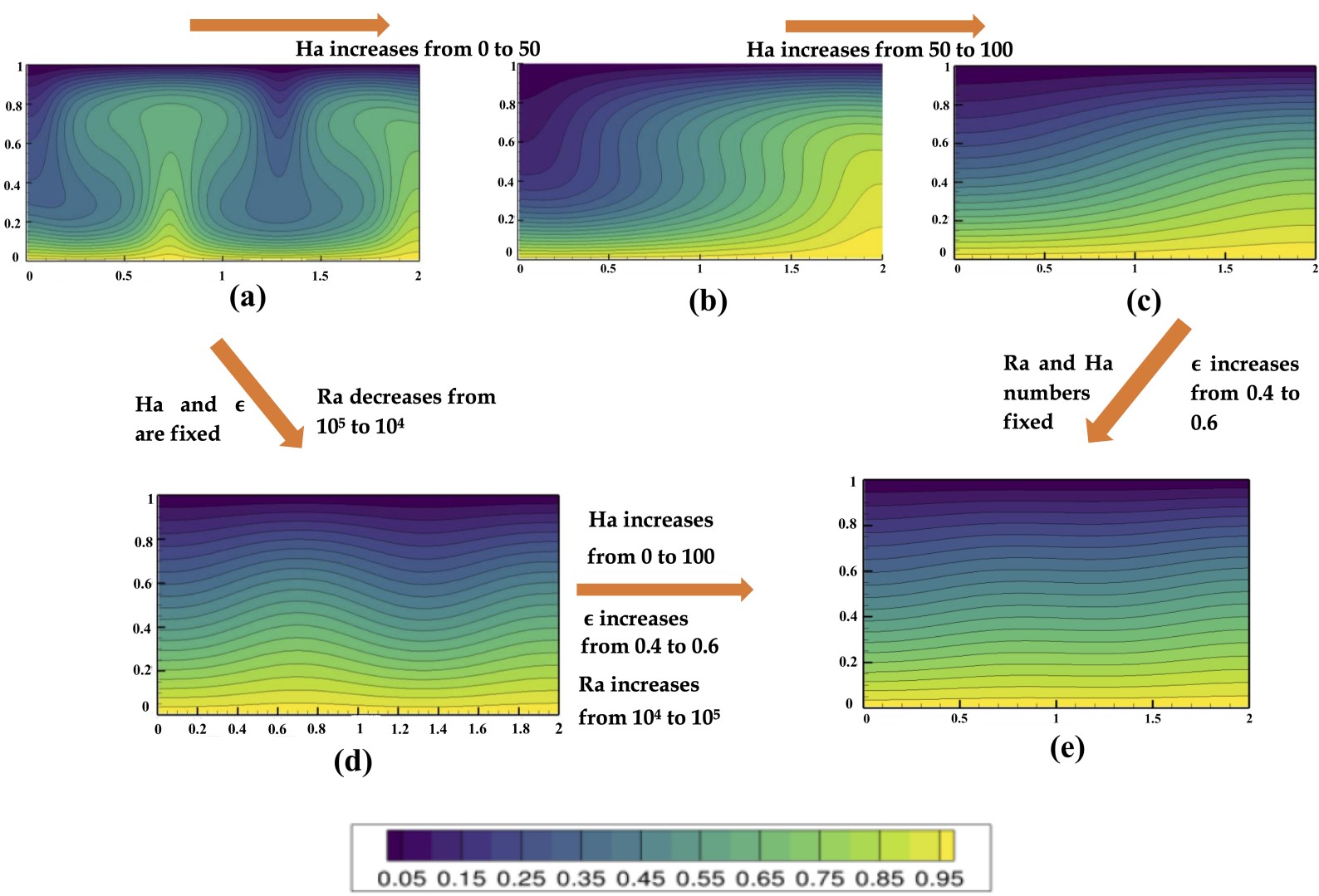

**Figure 4.** Representation of isotherms as *Ra*, *Ha* numbers, and $\epsilon$ changed under constant $\theta = 0$, and $Da = 10^{-2}$: (**a**) $Ra = 10^5$, $\epsilon = 0.4$, and $Ha = 0$, (**b**) $Ra = 10^5$, $\epsilon = 0.4$, and $Ha = 50$, (**c**) $Ra = 10^5$, $\epsilon = 0.4$, and $Ha = 100$, (**d**) $Ra = 10^4$, $\epsilon = 0.4$, and $Ha = 0$, (**e**) $Ra = 10^5$, $\epsilon = 0.6$, and $Ha = 100$.

*5.3. Predicting $\overline{Nu}$ from Ha Number and $\theta$*

In this part of this study, individual equations to predict $\overline{Nu}$ under the influence of an external magnetic field in an inclined rectangular cavity are developed for $Ra = 10^4$ and $Ra = 10^5$. The key element of this analysis is the consideration of the electrically conducting fluid in RB convection. Different simulations were conducted at $Ha$ number $\in [0, 100]$ based on the LBM model at $\theta \in [0, 90]$. The LM analysis was performed to build the correlation, followed by the validation with well-cited literature from the past and the recent. In general, good accuracy was established. The correlation is only valid for $Ra = 10^4, 10^5$, which are the most preferred ones for laminar flows as per the data from the literature.

5.3.1. Development of Correlation and Surface Analysis

Non-linear parabolic and power correlations were found to be the best-fitting ones among different functions for $Ra = 10^4$ and $Ra = 10^5$, respectively, under the LM algorithm. The correlation coefficients ($R^2$) were found to be 0.89 and 0.966 for $Ra = 10^4$ and $Ra = 10^5$, respectively. Figure 5 depicts the 3D contours for better visualization. It could be observed that the high density of the points was more aligned with low $\theta$, as most of the important transition in the heat transfer takes place under low $Ha$ and $\theta$. This behavior could be attributed to the effect of the magnetic field, which is directly controlled by the $Ha$ number. At an increasing $Ha$ number, the rate of heat transfer declines due to the existence of both an electric field and magnetic field, leading to the presence of a Lorentz force. Consequently, increasing the $Ha$ number lowers the values of $\overline{Nu}$. However, as part of the validation, a wide range of $Ha$ numbers and $\theta$ was considered to demonstrate the accuracy of the model and its ability to predict the heat transfer value outside the calibration zone.

As presented in Figure 5, $\overline{Nu}$ between 0.1140 and 5.280 was obtained from the surface analysis. The equation, however, is expected to be valid to predict $\overline{Nu}$ beyond the obtained range in the analysis due to the consideration of a broader range of $Ha$ and $\theta$. In order to obtain the equation from the best-fitting simulated contour from the LM algorithm, the LBM-simulated data were subject to several surface analyses for the purpose of interpolation within the user-defined range, and the following equation provided the best $R^2$:

$$\overline{Nu} = f + a\,exp(-Ha/b)exp(-\theta/c) \tag{36}$$

where $f$, $a$, $b$, and $c$ are fitting parameters with assigned values specifically under the aforementioned condition. Table 1 contains the values of the fitting parameters obtained through the LM algorithm with the best $R^2$ value.

**Table 1.** Fitting parameters to predict $\overline{Nu}$ from $Ha$ number and inclination angle $\theta$ at $Ra = 10^4$.

| Empirical Parameters | Fitting Values |
|:---:|:---:|
| $f$ | 0.87404 |
| $a$ | 1.73303 |
| $b$ | 30.0595 |
| $c$ | 127.50 |
| **Statistical Accuracy Indicators** | **Values** |
| $R^2$ | 0.966 |
| $p$ | $< 0.05$ |

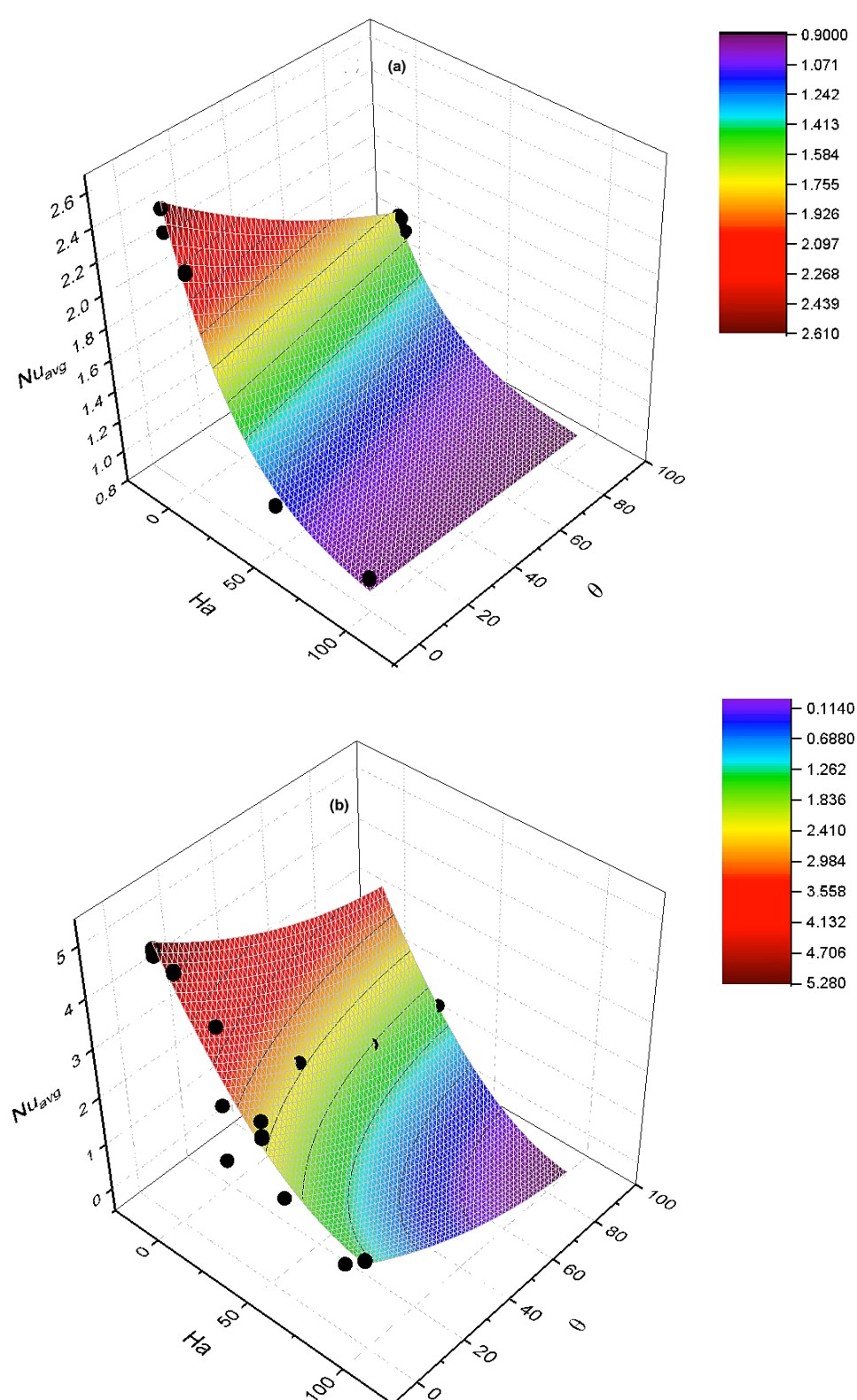

**Figure 5.** Development of correlation through fitting surfaces for (**a**) $Ra = 10^4$, and (**b**) $Ra = 10^5$ under the influence of external magnetic field at different inclination angles for electrically conducting fluid.

As $Ra$ increases from $10^4$ to $10^5$, the impact of buoyancy inside the enclosure augments significantly. Therefore, Equation (36) is not an appropriate option to predict $\overline{Nu}$ as a function of $Ha$ numbers and $\theta$. In fact, different functions were considered to determine the best-fitting surface to obtain the equation to predict $\overline{Nu}$ at $Ra = 10^5$. The following equation provided the best coefficient of determination to predict $\overline{Nu}$:

$$\overline{Nu} = f + aHa + b\theta + c(Ha)^2 + d(\theta^2) \tag{37}$$

**Table 2.** Fitting parameters to predict $\overline{Nu}$ from $Ha$ number and inclination angle $\theta$ at $Ra = 10^5$.

| Empirical Parameters | Fitting Values |
|:---:|:---:|
| $f$ | 5.26782 |
| $a$ | $-0.07321$ |
| $b$ | $-0.03499$ |
| $c$ | 3.43333 |
| $d$ | 0.000162 |
| **Statistical Accuracy Indicators** | **Values** |
| $R^2$ | 0.897 |
| $p$ | <0.05 |

It should be mentioned here that the $p$-value outlines the significance of the statistical study implemented in this section. The lowest $p$-value indicates that the null hypothesis was rejected, and the correlation is statistically significant. Overall, $p < 0.05$ was considered to be a good indicator to validate the model's accuracy.

### 5.3.2. Cross-Validation with Literature

The cross-validation was conducted with the literature with a similar objective. However, none of the literature provided any clear mathematical correlation among the parameters. The data from the literature were not considered to calibrate the LM model. Hence, the cross-validation serves as an independent validation to show unbiased agreement with the LBM and LM data within the considered range of input parameters.

The validation plots presented in Figure 6 demonstrate good agreement between $\overline{Nu}$ predicted from LBM and LM simulations. The majority of the points were obtained to be within the $\pm 5\%$ error lines. To build the model, the validation dataset contained a similar geometry considered in this study. The separate validation plots represent the agreement for two different $Ra$ numbers ($Ra = 10^4, 10^5$) considered in developing Equations (36) and (37). The empirical parameters presented in Tables 1 and 2 were considered to obtain the $\overline{Nu}$ as presented in Figure 6a,b, respectively. A separate Table 3 is presented to indicate the accuracy individually with the literature data considered for the validation. As mentioned in the caption of Table 3, some outliers were ignored in the individual $R^2$ calculation since it was already considered for the overall $R^2$ determination. It should be mentioned that filtering the outlier point is a common practice in statistical analysis, and hence the influential negative point can be ignored.

**Table 3.** Obtained $R^2$ in each validation with literature individually. *Detected outliers were removed for the correlation development.*

| Ra | Rudraiah et al. [59] | Kefayati [60] | Sheikholeslami et al. [61] | Sajjadi et al. [62] | Ahmed et al. [16] |
|:---:|:---:|:---:|:---:|:---:|:---:|
| $10^4$ | - | - | 0.9789 | 0.963 | 0.9789 |
| $10^5$ | 0.9862 | 0.9878 | 0.9662 | 0.967 | 0.9858 |

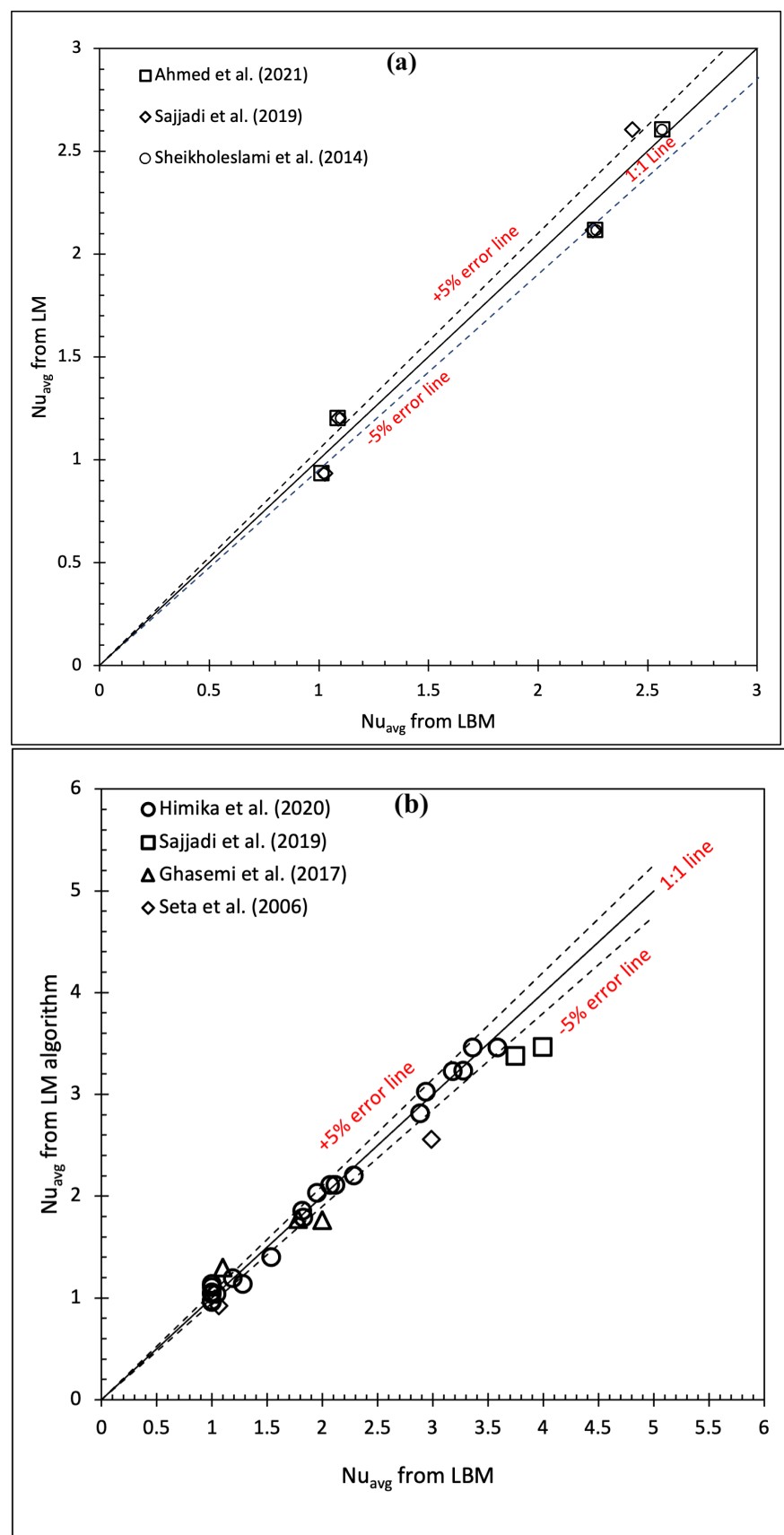

**Figure 6.** Cross validations with LBM data from the literature (**a**) $Ra = 10^4$ [16,61,62], and (**b**) $Ra = 10^5$ [18,62–64] under the influence of external magnetic field at different inclination angles for electrically conducting fluid.

### 5.4. Correlations among $\overline{Nu}$, Ha Numbers, and Da Numbers Under Constant Porosity

The major focus of this section is to predict $\overline{Nu}$ as a function of the $Da$ number ($0.1 \leq Da \leq 0.0001$) and the $Ha$ number ($Ha \leq 30$), at constant variables, such as porosity ($\epsilon = 0.4$) and inclination angle ($\theta = 45$). The data were obtained through LBM RB simulation, and a correlation was developed through the LM algorithm. Two different $Ra$ numbers $Ra = 10^4, 10^5$ were considered in this section of the study.

#### 5.4.1. 3D Fitting Curves and Statistical Parameters

Repeated LM algorithm simulations were performed to obtain the best-fitting results. As per Figure 7, the fitting curves are presented for $Ra = 10^4$ (Figure 7a) and $Ra = 10^5$ (Figure 7b), where the distribution of the LBM-obtained data is shown. At higher $Ha$ numbers, it was anticipated to have the lower $\overline{Nu}$; therefore, $Ha \leq 30$ was considered for the model calibration. However, the independent validation was conducted with the published literature, where the $Ha$ number was expanded up to 50, and still, significant agreement was established. On the other hand, a wide range of $Da$ numbers was considered in this part of the simulation. Therefore, the model could still develop a correlation at a higher $Da$ number.

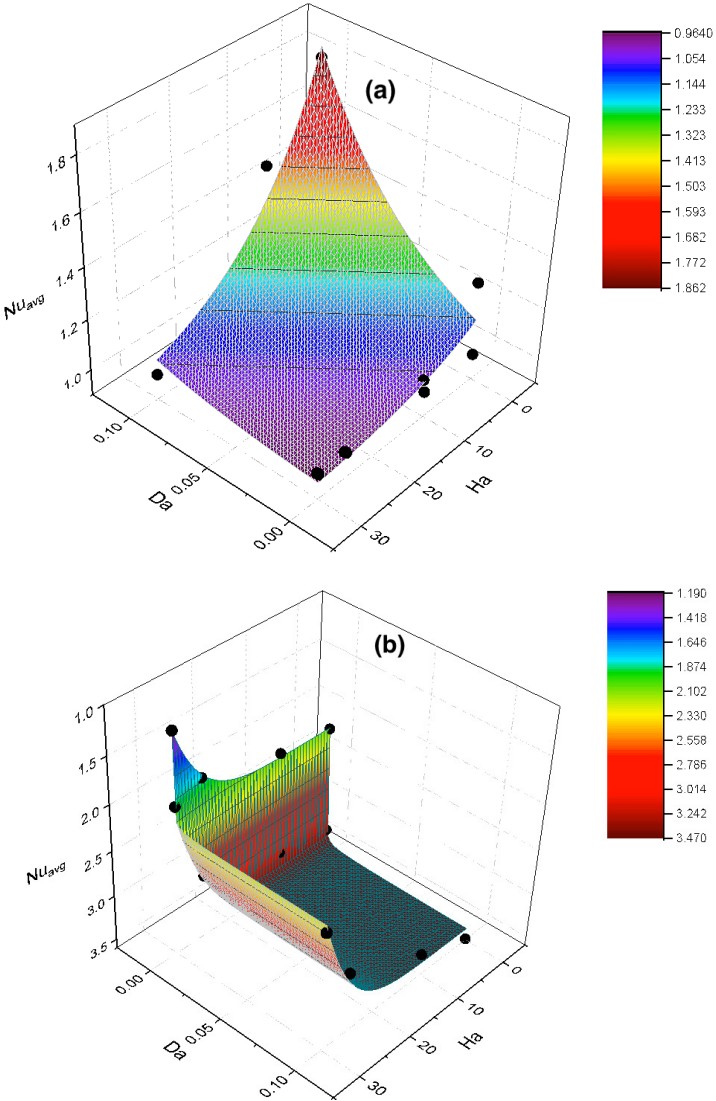

**Figure 7.** Independent validation plots by comparing with LBM data: (**a**) $Ra = 10^4$, and (**b**) $Ra = 10^5$, while $\epsilon = 0.4, \theta = 45$ were constant.

The obtained equation to predict $\overline{Nu}$ is the following (for $Ra = 10^4$):

$$\overline{Nu} = f + a\,exp(-Da/b)\,exp(-Ha/c) \tag{38}$$

The equation to predict $\overline{Nu}$ at $Ra = 10^5$ yielded the best $R^2$ under the power function, which was found as the following:

$$\overline{Nu} = f + a^{bDa} + c^{dHa} + e^{bdDaHa} \tag{39}$$

where $f$, $a$, $b$, $c$, $d$, and $e$ are empirical parameters to adjust the fitting surface and build the correlation. Tables 4 and 5 show the quantitative values of those parameters and statistical information of the LM model, which were the foundations of obtaining Equations (38) and (39).

**Table 4.** Quantitative values of empirical parameters to predict $\overline{Nu}$ from $Ha$ and $Da$ numbers at $\epsilon = 0.4$, $\theta = 45$, and $Ra = 10^4$.

| Empirical Parameters | Fitting Values |
|---|---|
| $f$ | 0.93997 |
| $a$ | 0.19833 |
| $b$ | $-0.06512$ |
| $c$ | 14.6836 |
| **Statistical Accuracy Indicators** | **Values** |
| $R^2$ | 0.90 |
| $p$ | $< 0.05$ |

According to Tables 4 and 5, the numerical values obtained from the LM algorithm are presented. The correlations were obtained to be $R^2 = 0.9$ and $R^2 = 0.99$, for $Ra = 10^4$ and $Ra = 10^5$, respectively.

**Table 5.** Quantitative values of empirical parameters to predict $\overline{Nu}$ from $Ha$ and $Da$ numbers at $\epsilon = 0.4$, $\theta = 45$, and $Ra = 10^5$.

| Empirical Parameters | Fitting Values |
|---|---|
| $f$ | 3.47307 |
| $a$ | $-0.00128$ |
| $b$ | $-0.75661$ |
| $c$ | $-3.4519 \times 10^{-9}$ |
| $d$ | 5.79634 |
| $e$ | $8.81812 \times 10^{-13}$ |
| **Statistical Accuracy Indicators** | **Values** |
| $R^2$ | 0.99 |
| $p$ | $< 0.05$ |

### 5.4.2. Independent Validation

The independent validation was conducted with the literature to showcase the ability of the correlation with data. The purpose of such an approach is to validate the present approach with well-cited data from the past and the recent, collectively.

Figure 8 presents the validation results obtained through the present approach for $Ra = 10^4$ (Figure 8a) and $Ra = 10^5$ (Figure 8b), respectively. It was found that the majority of the points were near the 1:1 line, and the agreement was within the $\pm 5\%$ error lines. The agreement plot also demonstrated the range of the $\overline{Nu}$ being as high as 4, which is mostly observed at a lower or no $Ha$ number. Therefore, the present approach was able to predict LBM results within multifarious ranges with a low percentage of error. The $R^2$ of each comparison in the validation is presented in Table 6, where $0.9113 \le R^2 \le 0.9928$ was obtained, which provides more confidence in the accuracy of the present approach. Due to

the limited data in the literature, the number of points presented in Figure 8a is less than those of Figure 8b.

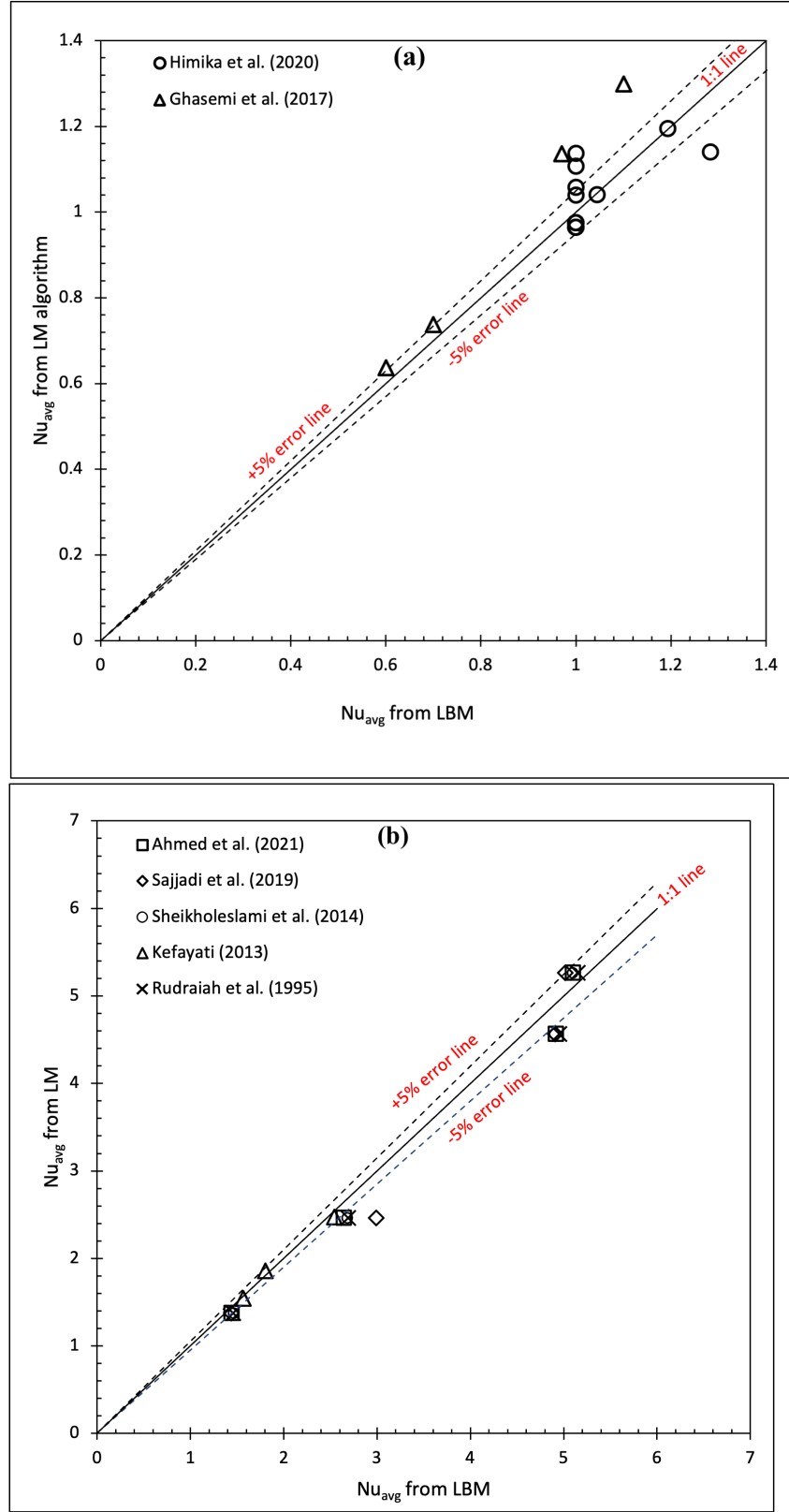

**Figure 8.** Surface fitting curves obtained from LBM data by LM algorithm for (**a**) $Ra = 10^4$ [18,64], and (**b**) $Ra = 10^5$ [16,44,59,61,62], while $\epsilon = 0.4$, $\theta = 45$ were constant.

**Table 6.** Obtained $R^2$ in each validation segment.

| Ra | Seta et al. [63] | Ghasemi et al. [64] | Sajjadi et al. [62] | Himika et al. [18] |
|----|------------------|----------------------|----------------------|---------------------|
| $10^4$ | - | 0.9946 | 0.9113 | - |
| $10^5$ | 0.9994 | 0.9362 | 0.967 | 0.9928 |

*5.5. Equation to Predict $\overline{Nu}$ under Variable Porosity*

In this segment of statistical analysis, porosity ($\epsilon$) is considered as a variable under the fixed $\theta$, and $Ra$ numbers. The primary focus is to establish a correlation under variable porosity ($\epsilon$), which is quite sensitive to other variables concurrently. Therefore, $Ra = 10^5$ and $Da = 10^{-1}$ are considered for the sensitivity analysis.

5.5.1. 3D Fitting over a Planar Surface

It was anticipated that under constant $Da$ and $Ra$ numbers, and $\theta$, the increasing rate of $\overline{Nu}$ will be linear as a function of $\epsilon$ and $Ha$, considering the fact that $Ha$ remains constant in each step while $\epsilon$ varies. For example, it was pinpointed earlier that the increasing $Ha$ number significantly plummets the heat transfer rate. However, if $Ha$ remains unchanged, yet $\epsilon$ increases, the $\overline{Nu}$ will increase linearly due to the improved convection inside the cavity since the $Da$ number is also unchanged.

Figure 9 serves as a testimony of the aforementioned statement, where a planar correlation was obtained through the LM algorithm. The $R^2 = 0.91$ depicts the accuracy of the correlation, which can be improved further with more relevant data within the plane. The distribution of the points within the surface implies that a suitable range was taken into account to predict $\overline{Nu}$ over the multifarious $0.4 \leq \epsilon \leq 1.0$ and $0 \leq Ha \leq 50$.

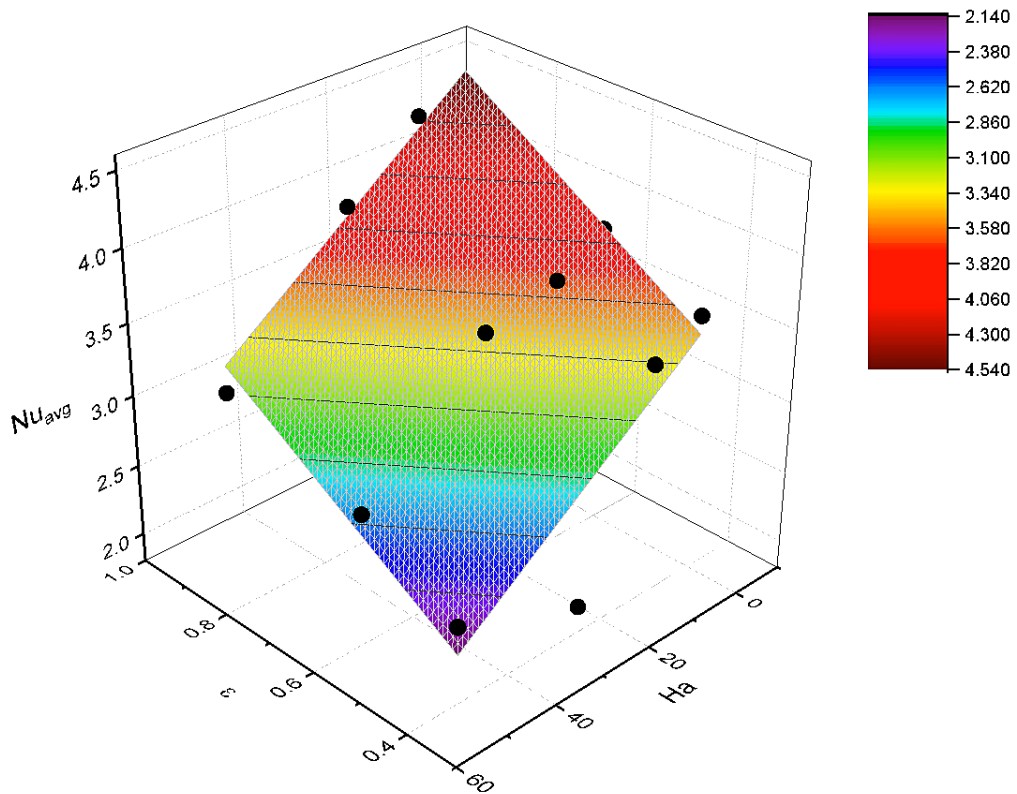

**Figure 9.** Fitting parametric development over a planar surface to predict $\overline{Nu}$ under constant $Ra = 10^5$, $\theta = 60$, and $Da = 10^{-1}$.

The equation to predict $\overline{Nu}$ under this circumstance was obtained to be the following:

$$\overline{Nu} = f + a\epsilon + bHa \tag{40}$$

The values of the empirical parameters from Equation (40) are presented in Table 7. The simplified version of the equation was also found to be effective at elevated $Ha$ numbers; however, due to the insufficient data in the literature, the validation was not conducted beyond $Ha = 50$.

**Table 7.** Parametric values to predict $\overline{Nu}$ from Equation (40).

| Empirical Parameters | Fitting Values |
| :---: | :---: |
| $f$ | 2.6105 |
| $a$ | 2.13773 |
| $b$ | $-0.02639$ |
| **Statistical Accuracy Indicators** | **Values** |
| $R^2$ | 0.91 |
| $p$ | <0.05 |

5.5.2. Validation Result

The immediate validation was conducted to assess the accuracy of Equation (40). The independent validation data were obtained from the literature reported by Himika et al. [18] and Sajjadi et al. [62]. The $\pm5\%$ error lines were also included in a similar manner for better visualization of the agreement.

Figure 10 illustrates the agreement between the literature (LBM data) and the present LM method. In general, most of the points were found within the error lines. The statistical accuracy indicators presented in Table 7 suggest that the agreement was still acceptable. The range of $\overline{Nu}$ was found to be close to 5 (LBM-obtained result), which was, in fact, obtained at $Ha = 0$ and the highest porosity considered in this research pipeline, i.e., $\epsilon = 0.9$.

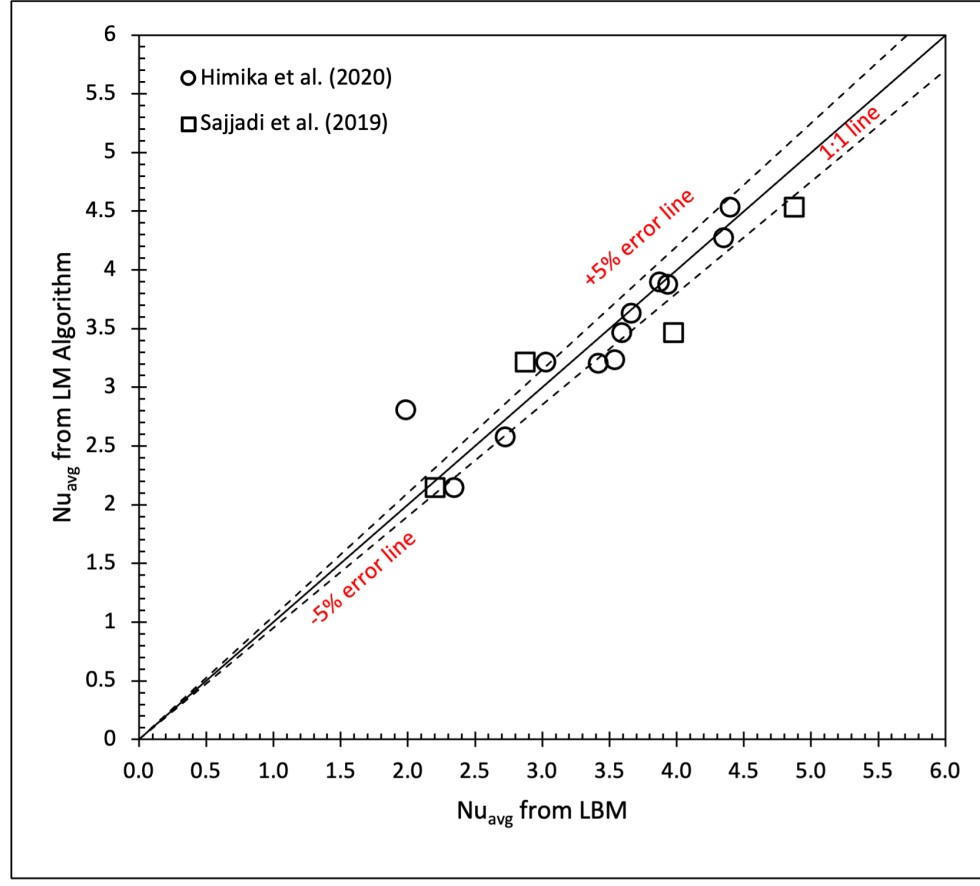

**Figure 10.** Fitting parametric development over a planar surface to predict $\overline{Nu}$ under constant $Ra = 10^5$, $\theta = 60$, and $Da = 10^{-1}$ [18,62].

Table 8 shows the $R^2$ values found from the agreement presented in Figure 10. The values of $R^2$ were found between 0.85 and 0.93 in comparison with the literature data mentioned earlier. All the relevant points were considered for validation. The specific point located beyond the $+5\%$ error line was still considered, and hence the $R^2$ was found to be 0.85 when comparing with Himika et al. [18]. However, if those specific data were left out, the $R^2$ increased significantly to 0.92.

**Table 8.** Coefficient correlations ($R^2$) obtained in comparison with individual literature as presented in Figure 10.

| Sajjadi et al. [62] | Himika et al. [18] |
|---|---|
| 0.93 | 0.85 |

## 6. Discussion

### 6.1. Significance of the Study

The LBM is a powerful and efficient alternative to solving fluid dynamics problems. LBM simulations can provide an accurate outcome within a shorter time scale compared to other relevant techniques, such as finite difference, finite element, and finite volume. All of the methods are accurate and have pros/cons. However, the explanation of the correlation among the input variables to quantitatively predict heat transfer was hardly reported. The present study utilizes input variables, such as $Ra$ number, $Da$ number, $\epsilon$, $Ha$ number, and $\theta$ to predict MHD-RB convection within a porous enclosure. The input parameters in the relevant study were mostly chosen to showcase the impact of buoyancy or porosity for the purpose of visualization. Nevertheless, there was always a gap in having equations numerically predict $\overline{Nu}$, which can be further validated with well-cited literature.

LBM data are highly non-linear. The current study considered the LM algorithm as one of the ML training methods to build non-linear surface analyses to develop three-dimensional correlation among the variables to predict $\overline{Nu}$. In each segment of the correlation development, validations were conducted, and statistical parameters were mentioned as part of the accuracy indicators. Furthermore, the empirical parametric values are provided in a tabular form after the statistical analyses, which will allow the researchers to reproduce the work based on the requirements. The reproducibility of the work by the LM algorithm could be established in different ways. One of them could be to perform direct LBM simulations by varying the input parameters to build the dataset, followed by correlation development to predict $\overline{Nu}$. In that case, the process could be time-consuming. The dataset could be split into 80% for the model development, and the rest 20% for the validation. However, validations with relevant well-cited literature need to be performed to understand the efficacy of the approach. It is recommended to consider a sample size of 100 for this approach. Alternatively, the interpolation of a limited dataset could be another option. However, this part also requires the initial model development and creating a dataset through simulations. However, the approach should consider the highest and lowest possible range of each parameter within the geometry to interpolate. The LM algorithm combined with the R library package *"dplyr"* assists the data manipulation without any involving any complicated steps. It is possible to establish a dataset with 3000 points through interpolation without the need to run LBM simulations varying each parameter. The present study considered both aforementioned approaches.

The current study aims to fulfill such requirements by interpolating datasets for the purpose of validations. The validations by comparing with identical geometries and physical properties with relevant literature provide another form of evidence on a high level of accuracy of the present approach. Some of the similar approaches in fluid dynamics study are worth mentioning. Recently, Islam et al. [6] presented correlations to predict $\overline{Nu}$ by considering input variables, such as Darcy ($Da$) numbers, Rayleigh ($Ra$) numbers, and porosities to predict $\overline{Nu}$ by the LM algorithm. At the same time, the equation and associated empirical parameters were utilized to validate the GPU-optimized LBM model.

However, the work of Islam et al. [6] was fairly restricted to heat-transfer phenomena of the nanofluid without the influence of an external magnetic field or representative non-dimensional parameter, such as the *Ha* number. Meanwhile, the present approach did not consider any machine learning or deep learning approach for the data-driven approach, but the inclusion of artificial intelligence in fluid dynamics study has become widely popular recently and should represent the state-of-the-art approach. It can be stated that no approach can be considered a direct validation yardstick, as numerical methods are not 100% correct, but the concept of correlation development as an additional form of validation has been reported. For example, the ANN modeling of nanofluid under magnetic field influence by Shah et al. [42], Alqaed et al. [41], and He et al. [43] attest to the purpose of the data-driven approach being included in LBM simulations of fluid flow. Nevertheless, this is the first work which has provided validations in each segment to showcase the potential of the data-driven approach in fluid dynamics by correlations development through the LM algorithm without the need for implementing machine learning methods or high computational resources.

Experimental fluid dynamics is time-consuming, delicate, and expensive [65,66]. If the boundary conditions, simulated data, and final representative contours are not fully understood numerically, the experimental approaches could cost a fortune. In an industrial setup, it will always be beneficial if numerical modeling contains correlations that are repeatable and reproducible. Based on the equations, the researchers will be able to test and tune the parameters within the domain to obtain the best-performing model to meet the scientific requirements. Then the final model development can be established through the original fluid flow simulations with the best-fitting input parameters to save time and increase productivity as well as profitability. The equations presented in this study had coefficients of determination ($R^2$) between 0.85 and 0.99, which are within the standard acceptable range.

*6.2. Factors Affecting the Accuracy of the Equations*

The present study is based on LBM data. To build a proper correlation, a wide range of datasets is required. While the present study established the correlations with a large number of the dataset, those did not cover the whole domain of the input parameters due to a lack of data for the validation. For example, the *Ha* number could be as large as 200 or even more, which could reduce the $\overline{Nu} < 0.001$. The LM algorithm will not be a viable option for this approach. A machine learning or deep learning approach should be an appropriate method for such an option. Figure 5 correlations represented $0 \leq Ha \leq 100$, and while model calibration was feasible, there were no relevant data found in the literature for comparison. Furthermore, the present study considered laminar flow only. The *Ra* number could be more than $10^7$, and it can cover a wide range of turbulence through the increased buoyancy. Nevertheless, a lack of efficient data in the literature within the considered geometry restricted the present study to explore further options. Since machine learning or deep learning was not considered in this study, further extension of the input parameters was not explored. Some of the references had only 3–4 relevant points (for example, Figure 6a), and therefore, the limited data could have affected the accuracy of the respective equation. While the surface analyses were conducted based on interpolation of the dataset with at least 3000 data, more validation data would have improved the $R^2$ values of independent validations. For instance, Figure 8b contained 31 points for the independent validation, and $R^2 = 0.99$ was obtained by comparing with three of the references [18,62,63].

In addition, the LM algorithm is susceptible to noise and may downgrade the efficiency of the neural network. However, those impacts are most visible for complex geometries. Since the present study considers a 2D porous cavity, the error percentage was acceptable. The re-tuning of the parameter would have been more efficient by considering any step associated with supervised machine learning model development. However, machine learning models require more data to train and test and cannot provide any equation for the

direct implementation in the study. The data considered in this study were sufficient for the LM algorithm and correlations development. The good agreement with the independent validations attest to such a statement.

*6.3. Future Recommendations*

The key findings of this study offer a major hint that a machine learning model can be developed to train LBM data into a well-suited model. Therefore, building correlations to predict an output parameter by considering multiple input parameters to build multifarious LBM models is highly recommended and is under strong consideration by the authors. Furthermore, the present study aimed to provide equations under different input parameters to predict $\overline{Nu}$. In most of the fluid dynamics research, entropy generation is also determined to demonstrate the fluid irreversibility. Both heat transfer and fluid friction are responsible for fluid irreversibility, particularly for nanofluid study. The correlation development to predict total entropy will also lead to another highly correlated parameter, the Bejan number (*Be*), which is the ratio of the heat transfer irreversibility to total irreversibility due to heat transfer and fluid friction. Therefore, determining the total entropy generation is one of the major indices in heat transfer application and should require involving more input parameters. Considering the multifarious input parameters involved in affecting entropy generation, machine learning is a suitable method and will be further explored.

Finally, the present study was validated with the simulated results from CPU-based computing. Considering the multivariate model development discussed above, CPU-based computing will be tedious in terms of the machine learning approach. In addition, a 3D implementation will be required to replace the 2D model, which could also consider more complicated lattice models, such as *D3Q*15 and *D3Q*19, replacing the *D2Q*9 of the present study. The possible implications of GPU-based LBM simulations will significantly reduce the computational time and increase the efficiency of the model. Considering a parallel computing platform, such as the Compute Unified Device Architecture (CUDA), could be a better method for implementing the MHD-LBM hybrid machine learning model. Some of the relevant works minus the machine learning have been published by the authors [14,67]. Therefore, developing a machine learning model through GPU computing for LBM cross-validations, possibly in a 3D geometry, will be a milestone within LBM research.

## 7. Conclusions

This study developed an LBM-MHD data-driven method to numerically predict the average *Nu* number ($\overline{Nu}$) as a non-dimensional representative value of the average rate of heat transfer by the LM algorithm. The mathematical correlations to predict $\overline{Nu}$ by considering *Ha* numbers, *Ra* numbers, inclination angles (*θ*), *Da* numbers, and $\epsilon$ were explored, followed by validations with the literature. The coefficients of determinations were found within $0.85 \leq R^2 \leq 0.99$, and this provides compelling evidence for the accuracy of the equations. The streamlines and isotherms were also presented to visually demonstrate the impact of the above-mentioned numerical parameters on the heat transfer phenomena. The equations presented in this study could be taken into account to validate any existing LBM-MHD model which considers RB convection within a 2D rectangular porous cavity. More options could be explored by directly developing a machine learning model to add extra features within the LBM model to establish benchmark solutions, which are under strong consideration for future study.

**Author Contributions:** Conceptualization, T.A.H., M.F.H., M.M.M. and M.A.I.K.; methodology, T.A.H., M.F.H. and M.M.M.; software, T.A.H., M.F.H. and M.M.M.; validation, T.A.H. and M.F.H.; formal analysis, T.A.H., M.F.H., M.M.M. and M.A.I.K.; investigation, T.A.H. and M.F.H.; resources, T.A.H., M.F.H. and M.M.M.; data curation, T.A.H. and M.F.H.; writing—original draft preparation, T.A.H. and M.F.H.; writing—review and editing, M.M.M. and M.A.I.K.; visualization, T.A.H. and M.F.H.; supervision, M.M.M. and M.A.I.K.; project administration, M.M.M. and M.A.I.K.; funding

acquisition, M.M.M. and M.A.I.K. All authors have read and agreed to the published version of the manuscript.

**Funding:** (i) North South University, Bangladesh, Grant No. CTRG-22-SEPS-09 (ii) Ministry of Science and Technology, Bangladesh Government, Grant No. EAS/SRG-222427.

**Data Availability Statement:** Data available on request.

**Conflicts of Interest:** The authors declare no conflict of interest. The funders had no role in the design of the study; in the collection, analyses, or interpretation of data; in the writing of the manuscript; or in the decision to publish the results.

## Abbreviations

The following abbreviations are used in this manuscript, shown in the order that they appear in the texts:

| | |
|---|---|
| LBM | Lattice Boltzmann method |
| MHD | Magnetohydrodynamics |
| RB | Rayleigh–Bénard |
| LM | Levenberg–Marquardt |
| 2D | Two-dimensional |
| Ha | Hartmann |
| Ra | Rayleigh |
| Darcy | Da |
| ANN | Artificial Neural Network |
| Nu | Nusselt |
| HPC | High-performance computing |
| AI | Artificial intelligence |
| ML | Machine learning |
| NLSM | Nonlinear least squares minimization |
| TLBM | Thermal LBM |
| BGK | Bhatnagar–Gross–Krook |
| SRT | Single-relaxation times |
| DF | Distribution functions |

**Nomenclature**

English symbols

| | |
|---|---|
| $a, b, c, d, e, f$ | Fitting parameters for LM-obtained equations |
| $B$ | Magnitude of magnetic field |
| $C_i$ | Lattice speed |
| $C_s$ | Speed of sound |
| $\overline{d_k}$ | Expected outcome |
| $\overline{e_i}$ | Discrete velocities |
| $f_i$ | Distribution function for flow fields |
| $f_i^{eq}$ | Equilibrium distribution function |
| $F_i$ | Force terms |
| $F_i^M$ | Force term for MHD |
| $F_i^P$ | Force term for porous media |
| $F_i^b$ | Buoyancy term |
| $g_i$ | Distribution function for temperature fields |
| $g_y$ | Gravitational force acting in y-direction |
| $g_i^{eq}$ | Thermal equilibrium function |
| $H$ | Height of the cavity |
| $K$ | Permeability |
| $m$ | Lattice on the boundary |
| $N$ | Sample number |
| $n$ | Iteration index |

| | |
|---|---|
| $Nu$ | Nusselt number |
| $\overline{Nu}$ | Average Nusselt number |
| $t$ | Time |
| $\Delta t$ | Time interval |
| $T$ | Temperature |
| $\Delta T$ | Temperature difference |
| $T_c$ | Cold temperature |
| $T_h$ | Hot temperature |
| $v$ | Velocity component |
| $w_i$ | Solution of the interpolation |
| $x_k$ | Random example for the output network |
| $z$ | Number of anticipated outcome from the network |
| Greek symbols | |
| $\alpha$ | Thermal diffusivity |
| $\alpha'$ | Optimization field matrix |
| $\beta$ | Thermal expansion coefficient |
| $\epsilon$ | Porosity |
| $\lambda$ | Mean-squared network error |
| $\mu$ | Dynamic viscosity |
| $\nu$ | Kinematic viscosity |
| $\omega$ | Weighting factor |
| $\phi$ | Angle of inclination |
| $\psi$ | Either velocity or temperature in the convergence |
| $\rho$ | Fluid density |
| $\sigma$ | Electrical conductivity |
| $\theta$ | Dimensionless angle of inclination |
| $\zeta$ | Adjustment parameter in each iteration cycle |

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
