# Peer review of "LBM-MHD Data-Driven Approach to Predict Rayleigh–Bénard Convective Heat Transfer by Levenberg–Marquardt Algorithm"

_axioms, doi:10.3390/axioms12020199_

Round 1

Reviewer 1 Report

 This article reports on a numerical study of Rayleigh-Bénard (RB) convection in a porous cavity in the presence of an magnetic field oriented differently than gravity.

In principle the presented work fits in the scope “mathematical physics” part of the listed areas of interest for Axioms.

The article combines three things: the physical system, the Lattice Boltzmann method (LBM) for solving the transport equations of momentum and energy and the analysis of the results using a Levenberg-Marquardt (LM) algorithm. The three elements are not at all new by themselves, and the added value should come from the combination.

However after analysis of the article it is clear that the explanations and the analysis are rather poor and the article is not recommended for publication. (See comments below)

The example studied must be considered a very simple toy problem in magneto-hydrodynamics (MHD): here MHD of LB is studied only in 2D and for low Rayleigh number (Ra) leading to laminar flow. As shown by this article the numerical simulation of this is possible on a small computer. Recent literature on MHD of RB goes much further. See the recent articles by Wilson et al  (International Journal of Heat and Fluid Flow 49 (2014) 80–90) and by Kenjeres (International Journal of Heat and Fluid Flow 73 (2018) 270–297)

The authors consider the use of LM algorithm as efficient algorithm in the framework of machine learning (ML) and refer to ref. [51] for this. From the abstract of [51] reproduced below it is clear that it concerns an application of the LM algorithm using the concept of “neural neighborhood” in a neural network.

It is not clear whether the LM as it is applied in this work is similar than the LM as applied in Ref. [51]. The RB problem is explained in sections 3.1-3.4 and then section 3.5 abruptly starts with explanation what the LM algorithm is. What is said is a rather confusing explanation. Better explanations can be found on Wikipedia!

Section 3.5 also contains equations with new symbols that are not properly explained. In equation 32 the variables alpha and lambda appear. The explanation assumes previous knowledge on neural networks. But this is not linked to the RB problem at hand. Which variables are considered input of the ANN? Which variable are output? How many layers? How many neurons per layer. Figure 2 gives a flow chart where the LBM data are the input and the structure of LM is explained. It gives insufficient information on the details.

The section ends with the statement “Once the training data is achieved, correlation development can be initiated”. This is not clear. When the training data are properly used the parameters of the network can be given best possible values. In this process the LM algorithm can help. When the training is completed one can proceed to testing and application. The neural network after the training already gives the correlation between input and output.

To be clear this referee would like to suggest another reference that has a much clearer structure on how the LM exploiting neural network can be used to solve an engineering problem, namely

ANN-based estimator for distillation using Levenberg-Marquardt approach

By:

Singh, V (Singh, Vijander) ; Gupta, I (Gupta, Indra) ; Gupta, HO (Gupta, H. O.)

Engineering Applications of Artificial Intelligence

Volume20

Issue2

Page249-259

DOI10.1016/j.engappai.2006.06.017

PublishedMAR 2007

The authors should take this as an example to restructure their article so that it is clear, complete and understandable.

In the Section 5 results are presented. In each case the dependent variable Nusselt number is represented as function of two independent parameters.

The result of the LM method is a function such as Eq. (36) graphically represented in the figure. The statement preceding this equation is “The following equation was obtained through the LM algorithm”. Probably what is meant here is that the best values of the four parameters are found using LM, whereas the analytic expression was selected by the authors from a set of options.

Equation (37) has another functional form. The sentence above it is not clear. It says that “While the parabolic form was also considered, the exponential function provided the best R2 as the following?

It seems that the quadratic form was used and the parametric values in Table 2 are the optimal ones, in contradiction with the sentence above Equation (37).

The section 5.1.2 reports on a comparison of literature data on Nusselt number with the Nusselt number obtained in this work. It is mentioned that this can provide “independent validation to show unbiased agreement with the LBM and LM data”. This is a too strong conclusion since the literature data all come with their limitations. How good was the computational approach used in literature? It is quite intriguing that there are outliers in the literature data. This needs further investigation. E.g. is there any visible difference in flow field and temperature field between literature data and current study?

The section on streamlines, 5.4 comes as an extra development, independent of the optimization problem. It is nice to see this, but one would expect this, maybe in shorter form in an earlier section where the results of the LBM are displayed, before the neural network study starts.

The discussion in chapter 6 is far reaching but a bit overambitious.

6.1 on significance is not well explained. Providing missing insight, providing a new application of the LM algorithm, recommendation on how the data-driven approach can be used should be clearly discussed and without too many vague statements.

6.2 on accuracy discusses on need for wide range of datasets in qualitative manner. This has no meaning if not illustrated with concrete numbers: how many data would be needed to accurate?

6.3 on future recommendations is a strange mix.

The part on the entropy generation is not understandable, because the problem was not properly introduced. The use of GPU is an open door in general, but for the small type of problem considered in this article is not relevant. So it should be complemented with estimates on how larger the problem is in 3D compared to 2D and how the computational effort scales with the Rayleigh number.

The statement that the model used here can be “taken into account to validate any existing LBM-MHD model which considered RB convection within a 2D rectangular porous cavity”.

Assuming that those models are based on the same underlying physics and based on the same set of transport equations as presented in this article, those model would give the same result. So an example of where the differences between models is, would be helpful.

SLOPPY

A big problem with this article is that it is sloppy in many ways.

In figure 1 the y-axis is from hot to cold wall and the x-axis is parallel to them.

However the equations (28)-(29) are only correct if x-axis is from hot to cold wall and y-axis is paralle to them

At the lefthand side of (12), the symbol v should be the greek letter \nu

In (12) and (13) the symbol for speed of sound appears that is only explained on the next page.

The symbol Greek letter \omega in equation (19) seems to be identical to the letter w in Equation (23).

More generally, the symbols are not systematically explained; some remain unexplained. It would be better to have a list of symbols at the beginning of the article.

It is disturbing that the legend in the figures 4 and 6 provides reference numbers for each dataset that are not correct. (Fortunately the references can be found in the list of references based on the name of the author). But it is another example of how sloppy this article is.

This is also seen in qualitative comments such as “The discrepancy could be attributed to … from this part of correlation development which was in an acceptable range”. What is acceptable really depends on the application context. The desired accuracy is determined by considerations of scientific challenge, product quality, safety considerations etc depending on the context.

TERMINOLOGY

Terminology can be improved:

Example:

The terminology “correlation among the numerical variables Ra, Da, Ha, inclination angle and porosity to predict the average of heat transfer (Nu) by LM algorithm” is in appropriate. In this field of study the aim is to find a correlation for a dependent physical parameter Nu as function of a set of independent physical parameters Ra, Da, Ha etc. There is no correlation between the independent variables, because they are set at fixed values. There is only correlation between Nu and the independent variables. On the other hand this is a numerical study and the correlation coefficients refer to the level of agreement between the basic data and the proposed analytical expression them.

REFERENCE [51]

It is given as very important reference underlying this work, but the neural network aspect is not all elaborated here so that the whole work is very difficult to understand.

Has the method of 51 been directly used or in a modified form?

Ref 51:

Neighborhood based Levenberg-Marquardt algorithm for neural network training

By:

Lera, G (Lera, G) ; Pinzolas, M (Pinzolas, M)

Volume13

Issue5

Page1200-1203

Article NumberPII S1045-9227(02)05564-9

DOI10.1109/TNN.2002.1031951

PublishedSEP 2002

Abstract of Ref. 51:

Although the Levenberg-Marquardt (LM) algorithm has been extensively applied as a neural-network training method, it suffers from being very expensive, both in memory and number of operations required, when the network to be trained has a significant number of adaptive weights. In this paper, the behavior of a recently proposed variation of this algorithm is studied. This new method is based on the application of the concept of neural neighborhoods to the LM algorithm. It is shown that, by performing an LM step on a single neighborhood at each training iteration, not only significant savings in memory occupation and computing effort are obtained, but also, the overall performance of the LM method can be increased.

Author Response

Uploaded

Reviewer 2 Report

Manuscript ID: axioms-2169457

Title: LBM-MHD data-driven approach to predict Rayleigh-Bénard convective heat transfer by Levenberg–Marquardt algorithm

In this study, an LBM-MHD data-driven method is introduced to numerically predict the average heat transfer rate (Nu) by the LM algorithm. The authors should address the following comments before the final decision:

 Avoid collective citations [1-6][7-9][10-13][14-18][19-22][23-26][27-29][33-37]…;

·         Explain the novelty of the study in more detail.

·         The following papers related to the application of LBM in fluid flow and heat transfer are missing in the literature review: 10.1016/j.molliq.2020.114941; 10.22055/JACM.2019.29503.1605

·         Show hydrodynamic boundary conditions in Fig. 1.

·         There is a mistake in Fig. 8: Sajjadi et al. reference number is not [54]! There are same mistakes in other figures.

·         Table 8 is not necessary to indicate separately. I suggest to show this information in Fig. 8.

·         In conclusion: “Nu” refer to non-dimensional Nusselt number and it is different from average heat transfer rate.

·         Nomenclature should be added in order to define all variables and abbreviations used in the paper.

Author Response

uploaded

Reviewer 3 Report

The scientific article is made at a fairly high professional level. The research topic is not new, but the authors considered new aspects in their work. I have a few comments only on the design of the article for this scientific journal. I hope that the authors will correct these small comments without re-reviewing.

Author Response

Uploaded

Round 2

Reviewer 1 Report

The authors have made significant improvements compared to the first version. The general judgement of the work remains that it is an interesting field in mathematical physics. The unclear presentation of the method and the presence of disturbing inaccuracies in notation has been fully repaired in the revision. The article is now suitable for publication.

Reviewer 2 Report

All the comments are addressed.